# Parasitic-Aware Simulation-Based Optimization Design Tool for Current Steering VGAs

Nehad Mansour * , Mohamed Elnozahi and Hani Ragaai

Electronics and Communications Department, Faculty of Engineering, Ain Shams University, Cairo 11517, Egypt
* Correspondence: nehad.mansourr@gmail.com

**Abstract:** Designing millimeter-wave variable gain amplifiers (VGAs) is very challenging owing to the parasitic effects of the interconnects of both active and passive devices. An automated parasitic-aware optimization RF design tool is proposed in this paper to address this challenge. The proposed tool considers the parasitic effects prior to layout. It employs a knowledge-aware optimization technique. The augmentation between parasitic-aware and knowledge-aware techniques speeds up the design process and leads to a design as close to the final design after finalizing the layout. The proposed tool gives limitless and guaranteed converged solutions in a wide range of RF frequencies. A four-bits current steering VGA design is used as a validation of the tool. The tool is tested on three different frequencies using the 65 nm-technology node. The three tested frequencies (7, 10, and 13 GHz) show a root mean square gain error at approximately 0.1 dB and a phase variation at approximately 3.5° within a 16-dB gain control range. To our knowledge, it is the first reported automated design tool for a current steering VGA.

**Keywords:** design automation; knowledge-aware optimization; parasitic-aware design; parasitic modeling; current steering; digital control; low phase error; millimeter-wave (MMW); variable gain amplifier (VGA)

## 1. Introduction

Today, the telecom industry is rapidly expanding the deployment of mm-wave 5G technologies to meet the demand for higher data rate wireless signals. Phased-array beamforming is used to mitigate the range difficulty in these systems. Controlling the phase and magnitude of the signals at every antenna element creates constructive and destructive electromagnetic interference patterns over the air, generating physical, 3D beams. A variable gain amplifier (VGA) circuit is an essential building block that is responsible for controlling the gain of different streams for beamforming applications [1]. VGA is used to produce a range of gain states with minimal RMS gain step error smaller than 0.2–0.3 dB where this amplitude adjustment helps the phased array to achieve high sidelobe suppression, while maintaining the least possible RMS phase error between states.

Designing VGAs at mm-wave frequencies is challenging because of the parasitics that contribute to performance at these high frequencies. Electromagnetic simulations are necessary at these frequencies to capture the effect of wiring and coupling between the different passive components, such as on-chip inductors [2]. This increases the design cycle significantly and optimum design may not be achievable. Parasitic-aware design techniques help to speed up the design processes by estimating the parasitics of interconnects of both active and passive devices. In addition, automation and optimization of the design can reduce the design time as well as lead to an optimum design.

Design automation is mainly divided into two main categories, knowledge-based approach and optimization-based approach [1–5]. The main advantage of the former is the speed, while the disadvantage is the time needed to develop the knowledge database, done before the process. The latter is divided into sub-categories, equation-based optimization,

and simulation-based optimization [3]. The equation-based approach suffers from accuracy, especially if parasitics are included, while for the simulation-based approach a larger processing time is required [3]. An equation-based approach suffers from accuracy, especially if parasitics are included, while for the simulation-based approach a larger CPU time is required [3]. For both approaches, parasitic estimation during the optimization leads to a higher accuracy as well as fewer design iterations. A parameterized layout generator and parasitic extraction are used to consider the effect of on-chip inductors and interconnect parasitics shown in [4]. The parasitics are modeled with ideal components within the schematics of the circuit that is being optimized. Other approaches use approximate inductor analytical models, 2–π-models [5] and have been applied to different circuit classes. In [2], an algorithm that performs parasitic-aware automatic layout for analog/RF integrated circuits is presented. The algorithm creates a reduced-template-graph from original layouts and adds parasitic constraints. Using a two-dimensional hybrid scheme of graph-based optimization and nonlinear programming, the nonlinear problem is solved. The algorithm has successfully retargeted operational amplifiers and an RF low-noise amplifier within minutes of CPU time. An optimization methodology, presented in [6], based on adaptive simulated annealing (SA) with tunneling algorithm and a post-optimization PVT design centering strategy is used to model the parasitics of a self-biased fully differential RF CMOS PA. More recently another approach that combines genetic optimization algorithm and performance models (PMs) is presented in [7]. Device and interconnect parasitics are modeled into symbolic models, using foundry-provided equations and analytical models, respectively. Finally, to avoid in-the-loop EM simulations, in [8,9], a Pareto-optimal front (POF) of EM-simulated inductors is obtained prior to any circuit optimization, then, the POF is used as inductor design space (IDS) during circuit sizing. Another approach in [10] exploits the full capabilities of the most established computer-aided design tools for RF design available nowadays, i.e., RF circuit simulator as performance evaluator, electromagnetic simulator for inductor characterization, and layout extractor to determine the complete circuit layout parasitics. Liu et al. [11] proposed a simulation-based optimization approach for RF amplifiers, where machine learning techniques are used to build an inductor surrogate model. The accuracy of such a model is iteratively improved by refining the model with EM simulation results of promising inductors, instead of performing EM simulation of each candidate inductor, shown in [12].

A novel automated parasitic-aware simulation-based optimization design tool for designing mm-wave current steering VGAs is proposed in this paper. The augmentation between the parasitic-aware and knowledge-aware optimization methodologies speeds up the design process and help to achieve a design as close to the final design after finalizing the layout. To our knowledge, it is the first reported automated tool used to design current steering VGAs.

The paper is organized as follows: Section 2 presents the chosen architecture of the mm-wave digitally controlled current steering VGA. In Section 3, an overview of the proposed tool will be presented. Two subsections of Section 3 will address in detail the parasitic estimation methodology and the knowledge-aware simulation-based optimizer. The optimizer subsection explains the RF design trade-offs and the optimization flow. Section 4 presents the verification and simulation results, where the results of three test cases for the adopted VGA and the template layout are presented. Finally, in Section 5, conclusions are drawn.

## 2. N-Bits Digitally Controlled Current Steering Variable Gain Amplifier (VGA)

The most common topology of mm-wave VGAs relies on current steering using a cascode device [13,14]. Another circuit topology of VGA is to use an amplifier followed by a digital step attenuator [15–17], but the attenuator increases the circuit noise losses. Finally, current splitting techniques are another method to control the gain of the VGA [18].

Current steering topology is preferred over its alternatives due to its constant current and transconductance under different gain states, but the operating bandwidth would

be limited by the current steering circuits. This topology is usually implemented using differential architecture for enhanced linearity [19–22].

In this paper, the automatic sizing of the digitally controlled current steering VGA shown in Figure 1 is proposed. The VGA is similar to a differential amplifier with a cascode input stage [1]. Four control bits enable/disable a few of the cascode devices to steer the current away/to the load to control the gain. Transistors Mp1-4 are used to ensure that a constant current is flowing through the main transistor Mm. By correctly sizing the transistors (Mm, Mcas, Mp1-4, and Mc1-4) a linear-in decibel gain step can be achieved. Turning on the auxiliary transistors decreases gain but the output impedance changes, which results in phase variations between the different gain states. An RC feedback network ($R_F$ and $C_F$) is added to the cascode transistor for stability and wide bandwidth realization.

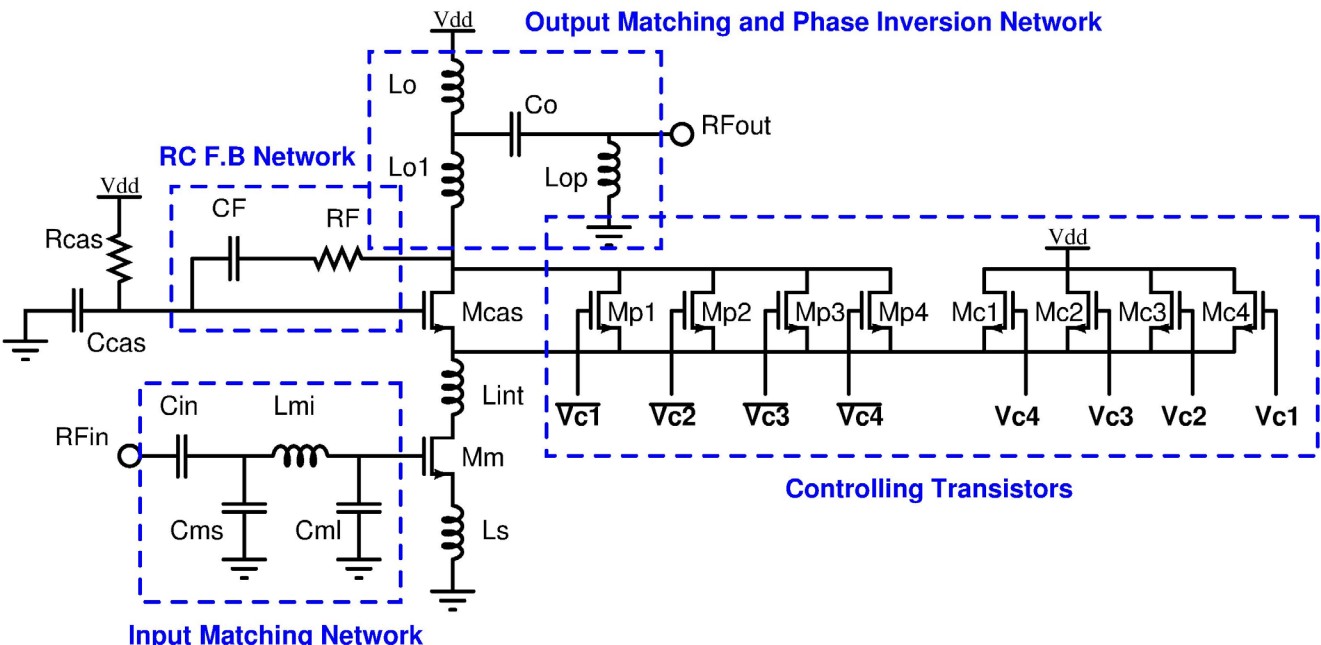

**Figure 1.** Circuit schematic of a single ended architecture for the proposed 4 bits digitally controlled VGA.

The cascode amplifier is conventionally used in broadband circuits to improve reverse isolation of the transistor. At high frequencies, design of cascode amplifiers entails resolving several issues one of them is the effect of parasitic inductive components associated with bias lines, interconnects, ground back-vias, bypass capacitors and bond-wires. This inductive effect results in cascode amplifiers instability at high frequencies. The stability can be improved by inserting a series gate resistance in the gate of the CG device and in some cases, it is needed to add capacitance. It has been shown that this RC network should be selected within the specific ranges to improve stability; otherwise, it can even degrade the amplifier stability [23].

Decreasing the root mean square (RMS) phase error is achieved by adjusting the values of the output matching network together with adjusting the interstage inductor to help tune out the equivalent parasitic capacitance at the intermediate node to minimize the phase/impedance variation during gain tuning. Source degeneration is added to improve linearity.

## 3. Proposed Automated Web-Based Design Tool

Figure 2 shows a top-level description of the proposed parasitic-aware design tool for the mm-wave digitally controlled current steering VGAs. The four main pillars of the presented tool are a web interface, a CAD RF circuit simulator, a simulator-based optimizer, and a parasitic device model generator. The proposed tool uses a parasitic device model

generator to model the parasitics of both active and passive devices such that the generated design is as close as possible to the final design after finalizing the layout.

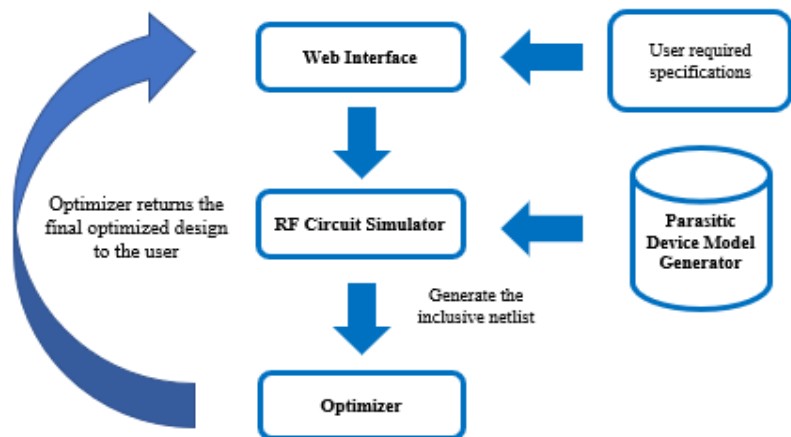

**Figure 2.** Top-level description of the proposed parasitic-aware design tool for the mm-wave VGAs controlled VGA.

The web interface is developed using PHP, HTML, CSS, and JavaScript. It is used to get the user's required specifications. Shell scripting is used to handle the logic of the design flow. It communicates with the RF simulator using ocean scripting and calls the optimizer to run the proposed optimization algorithm. Once the design is finalized, the final report is generated and displayed to the user through the web interface.

The parasitic device model generator estimates the parasitics to be used during the optimization phase to speed up the design time and reduce the iterations after generating the layout. Knowledge-aware simulation-based optimization is used to quickly size the circuit with a reduced number of iterations. In the following subsections, the parasitic device model generator and the knowledge-aware simulation-based optimizer are explained in detail.

### 3.1. Parasitic Device Model Generator

Modeling of inductors and capacitors includes lots of non-idealities. For inductors, ohmic losses, parasitic inductive effects, substrate effects, unknown ground return paths, and self-resonances are examples of those non-idealities. While for capacitors, those non-idealities include parasitic capacitance to ground, ohmic losses, and self-resonances [23]. For the proposed design automation tool, the models of capacitors and inductors provided by the foundry PDK are used directly because of the high accuracy of modeling.

Regarding the active devices, the proposed parasitic device model generator estimates the parasitics of the active device after extraction during the optimization loop. Those parasitic capacitances include the interconnections capacitance, extra gate-to-source capacitance $\left(C_{gs}\right)$, extra gate-to-drain capacitance $\left(C_{gd}\right)$, and extra drain-to-source capacitance $(C_{ds})$ introduced because of wiring. The model was generated using linear regression method to have an analytical expression for those capacitances. The linear equations for those extra capacitances are defined as follows:

$$C_{gs}(fF) = (-18.46 + 0.676\,N + 14.93\,w\,(\mu m))M, \tag{1}$$

$$C_{gd}(fF) = (-17.83 + 0.683\,N + 13.87\,w\,(\mu m))M, \tag{2}$$

$$C_{ds}(fF) = (-29.02 + 0.907\,N + 23.45\,w\,(\mu m))M, \tag{3}$$

where N is the number of fingers, M is the number of multipliers, and w is the width per finger in μm of the CMOS transistor. Figures 3 and 4 show a comparison of the extracted extra capacitance versus the one generated using (1)–(3) for different channel width and

assuming minimum channel length. As depicted, the equation models the extra capacitance with high accuracy. This analytical expression helps to speed up the optimization loop.

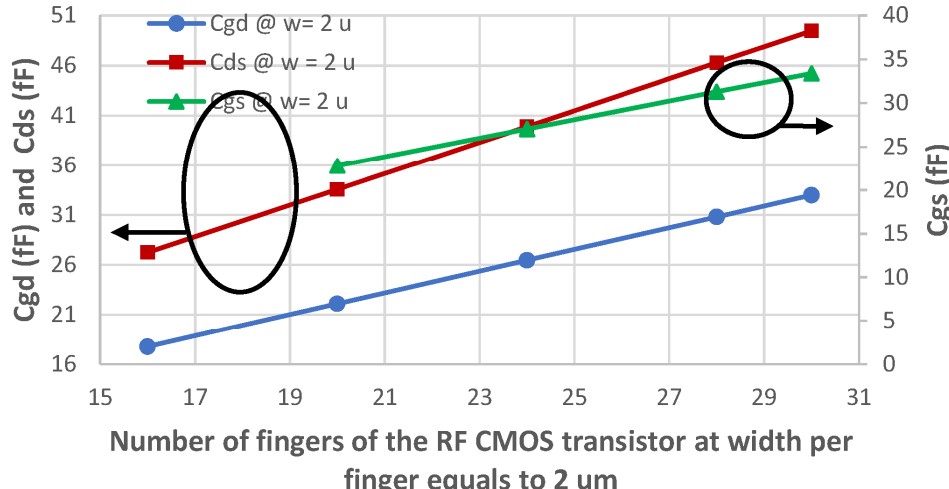

**Figure 3.** Estimated extra parasitic gate-to-drain capacitance, extra parasitic drain-to-source capacitance, and extra parasitic gate-to-source capacitance versus MOSFET's number of fingers (N) at width per finger equals 2 $\mu m$.

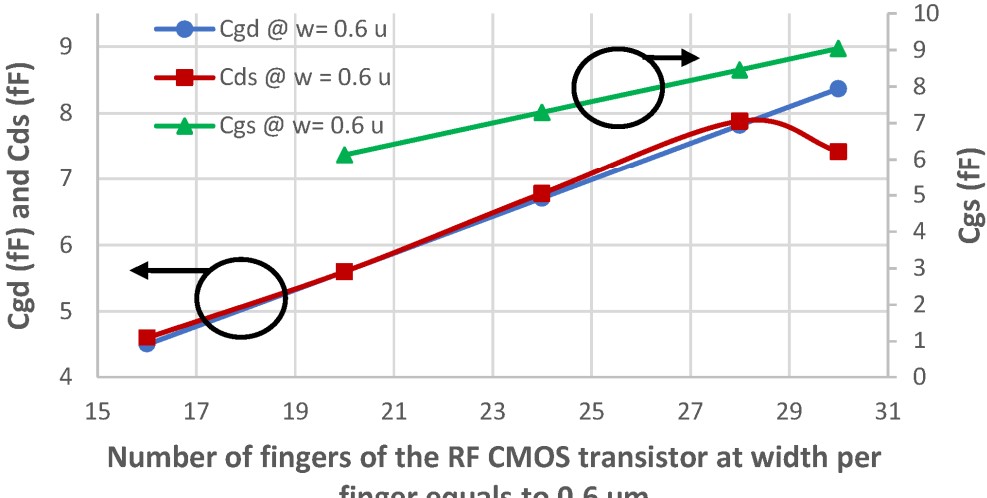

**Figure 4.** Estimated extra parasitic gate-to-drain capacitance, extra parasitic drain-to-source capacitance, and extra parasitic gate-to-source capacitance versus MOSFET's number of fingers (N) at width per finger equals 0.6 um.

### 3.2. Knowledge-Aware Simulation-Based Optimizer

RF design tradeoffs are very challenging, where the value of one design parameter could determine several of the design specifications. This could lead to a longer optimization time as well as the optimizer may not reach a solution. The proposed optimization algorithm is based on the knowledge of the designer by dividing one complex optimization into several smaller optimization steps to overcome the problem. Weighting factors are also used to determine which design specification is important to meet. The optimization core uses Broyden–Fletcher–Goldfarb–Shanno (BFGS) optimization algorithm implemented within the virtuoso environment [24]. The details of the knowledge-aware simulation-based optimization are discussed below starting from determining the different design specifications trade-offs and their dependency on the design parameters and ending with the proposed design flow.

### 3.2.1. Design Specifications Trade-Offs

The VGA, shown in Figure 1, has several design parameters. These design parameters determine the final specifications. The design specifications include input/output matching, gain, linearity, phase variation with step, etc. To avoid having time-consuming iterations during the optimization, the sensitivity of the design specification to the various design parameters is analyzed below. The sensitivity is determined using analytical equations and verified with simulations.

The input impedance of the VGA, shown in Figure 1, can be obtained using the simplified schematic in Figure 5. In the analysis below, the gate to source parasitic capacitance is considered. It could be shown that the input impedance is given by:

$$Z_{in} = \frac{s^2 C_{ml} L_{mi} Z_1 + s L_{mi} + Z_1}{1 + s C_{ml} Z_1 + s C_{ms} Z_1}$$

$$where, \quad Z_1 = \frac{1}{s C_{gs1}} + s L_s + \frac{g_{m1} L_s}{C_{gs1}}$$

(4)

where $s$ is Laplace variable s $(= j\omega)$, $L_{mi}$ (H) is the inductance of the input matching network, $C_{ms}$ and $C_{ml}$ are the capacitances of the input matching network, $C_{gs1}$ and $g_{m1}$ are the main transistor ($M_m$) gate-to-source parasitic capacitance and transconductance, respectively, and $L_s$ is the source degenerated inductance. As depicted in (4), the input impedance depends mainly on the input matching network parameters, the transconductance of the main transistor, and the source degenerated inductor, $L_s$.

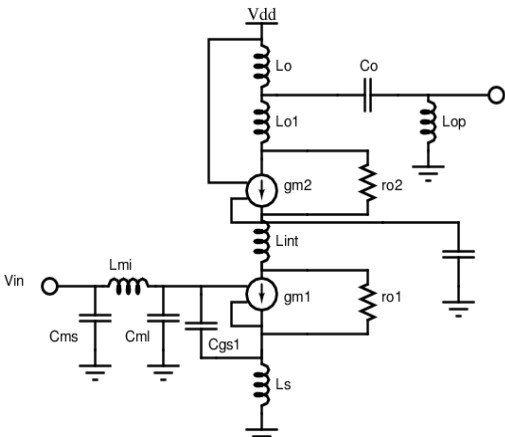

**Figure 5.** Model for the single VGA stage including the Cgs parasitic capacitance.

The max gain for the VGA is given by:

$$A_v = \frac{s g m_1 L_o}{(s L_o + \frac{1}{B}) \times (B) \times (1 + S g_{m1} L_s)}$$

where

$$B = \frac{1}{s L_{o1} + r_{o2}[g_{m2}(s g_{m1} L_{int} - 1)/(s^2 g_{m1} c_{gs2} L_{int} - s c_{gs2} + g_{m1}) + 1] + \frac{1}{s L_{op}}}$$

(5)

where $A_v$ is the overall gain, $s$ is Laplace variable s $(= j\omega)$, $gm_1$ and $gm_2$ are the main and the cascode transistor's transconductance, respectively, ($L_o$, $L_{o1}$, and $L_{op}$) are the inductors of the output matching network, $L_{int}$ is the interstage inductor between the main and the cascode transistors, $c_{gs2}$ is the parasitic gate-to-source capacitance of the cascode transistor, and $r_{o2}$ is the output resistance of the cascode amplifier.

In addition, the RMS gain error between its different steps can be found using the:

$$RMS\ gain\ error = \sqrt{\frac{1}{N-1} \sum_{n=2}^{N} |\Delta S21_i - S21_{step}|^2}$$

(6)

where N is the number of states, $\Delta S21_i$ is the step between the gain of state (i) and gain of state (i−1), and $S21_{step}$ is the achieved gain step between the N states.

Equation (5) indicates that the max gain depends on the output matching network parameters, the transistors transconductances, and the interstage inductor, and is inversely proportional to the source degenerated inductor. Furthermore, those parameters have a direct impact on the linearity (Input P1dB). Figure 6 shows the simulated gain, $S_{21}$, and the simulated IP1dB versus the value of the input matching inductor, $L_{mi}$. As depicted, the max gain is achieved for an inductance value of 500 pH, while the input P1dB reaches its minimum value. This shows the trade-off between those design specifications. During the optimization, the inductance of the input matching network, $L_{mi}$, should be considered during gain and input P1dB optimization.

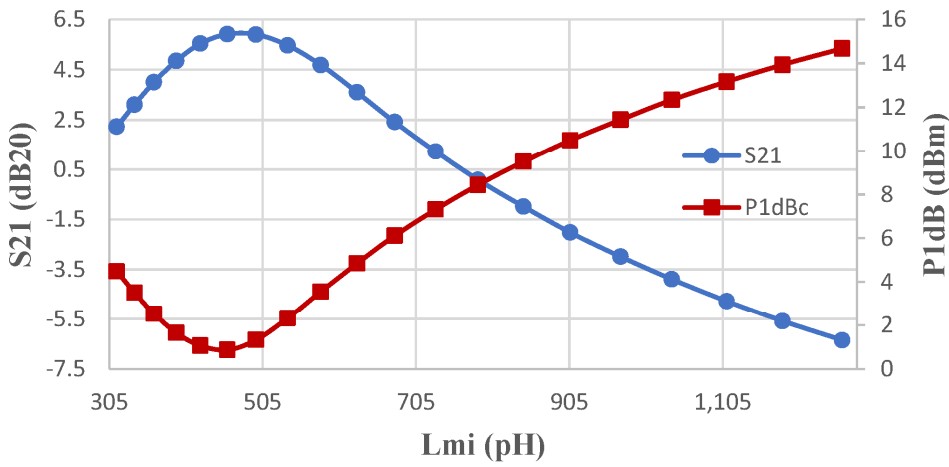

**Figure 6.** Gain and Input P1dB versus the input inductance of the input matching network.

Another trade-off between gain and the input P1dB appears versus current through the main device and it's sizing as shown in Figures 7 and 8. Simulations show another trade-off between gain and stability for the presented VGA architecture also seen in Figure 8. As depicted, increasing the main amplifier channel width ($W_m$) increases the gain of the VGA but decreases its stability factor ($K_f$).

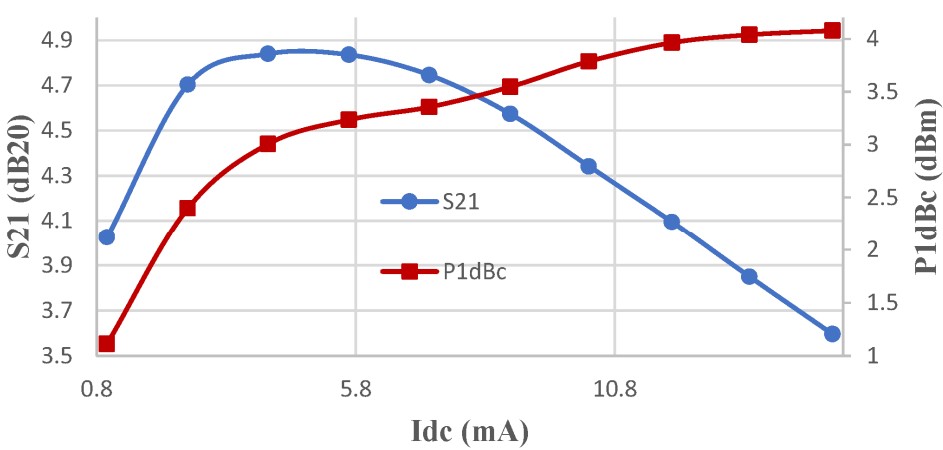

**Figure 7.** Gain and Input P1dB versus DC current source.

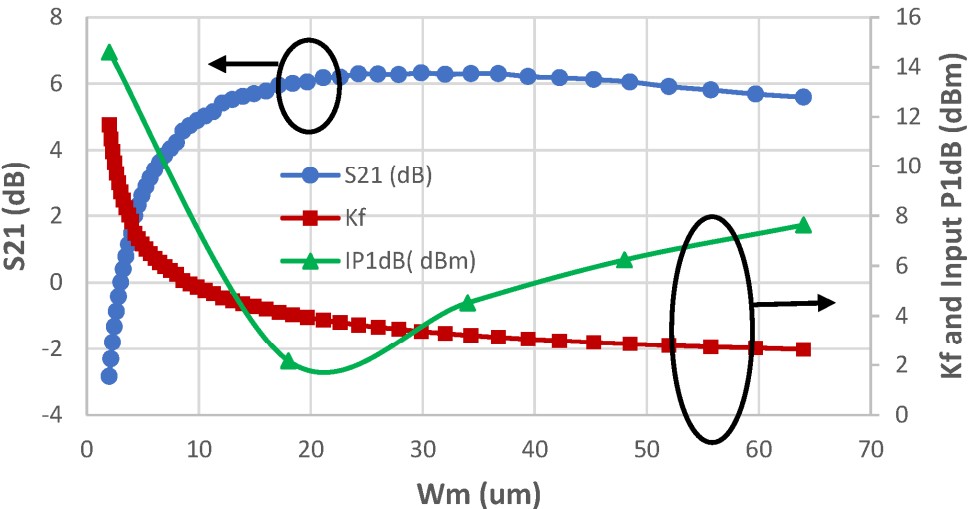

**Figure 8.** Gain, stability, and Input P1dB versus the main transistor channel width.

Phase variations between states are another important design specification at VGAs design, which is targeted to be minimal. The phase of every state is obtained using the:

$$
\begin{aligned}
\angle A_v = \tan^{-1} \frac{Imaginary\ (A_v)}{Real\ (A_v)} = \\
90° - \quad \tan^{-1} \left( \frac{\omega L_o + \omega L_{o1} - \frac{1}{\omega L_{op}} - \omega g m_2 g m_1^2 C_{gs2} L_{int}}{r_{o2}} \right) \\
- \tan^{-1} \left( \frac{\omega L_{o1} - \frac{1}{\omega L_{op}} - \omega g m_2 g m_1^2 C_{gs2} L_{int}}{r_{o2}} \right) \\
- \tan^{-1} (\omega g_{m1} L_s)
\end{aligned}
\tag{7}
$$

Some parameters which help in decreasing the phase variations between states to minimum also affect linearity. Figure 9 shows that increasing the output inductance increases the input P1dB but also increases the phase variations. This shows the trade-off between those design specifications. During the optimization, the inductance of the output matching network, $L_o$, should be considered during phase and input P1dB optimization.

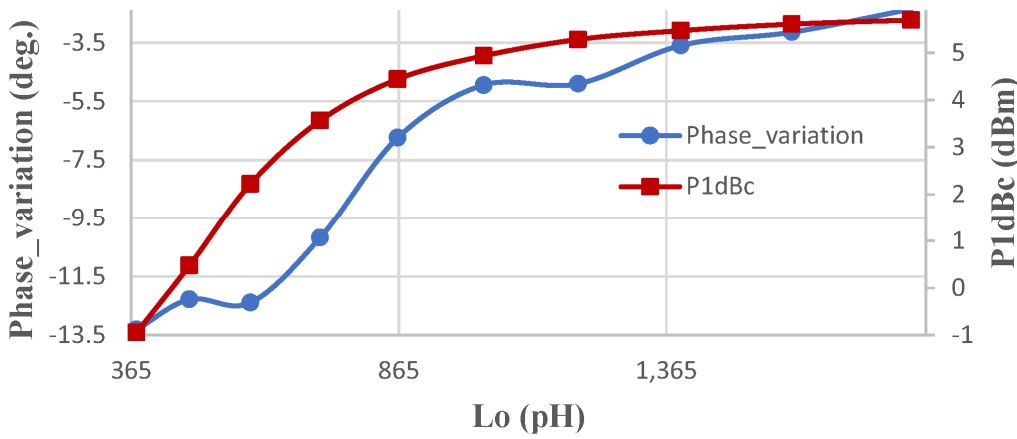

**Figure 9.** Phase variation and the input P1dB versus the output inductance.

The RMS phase error is given by:

$$
RMS\ phase\ error = \sqrt{\frac{1}{N} \sum_{n=1}^{N} |\varnothing_i - \varnothing_{i-1}|^2}
\tag{8}
$$

where N is the number of states, $\varnothing_i$ is the phase of state (*i*) and $\varnothing_{i-1}$ is the phase of state (*i*−1).

Figure 10 shows that well-sizing of the controlling transistors ($M_{C1-4}$) results in increasing the gain-step between states and decreasing the phase variations between them. Table 1 summarizes the main specifications with their corresponding controlling design parameters to better guide the optimization process. Together with the weighting factors methodology, the optimizer operates faster and its possibility to enter an infinite loop diminishes.

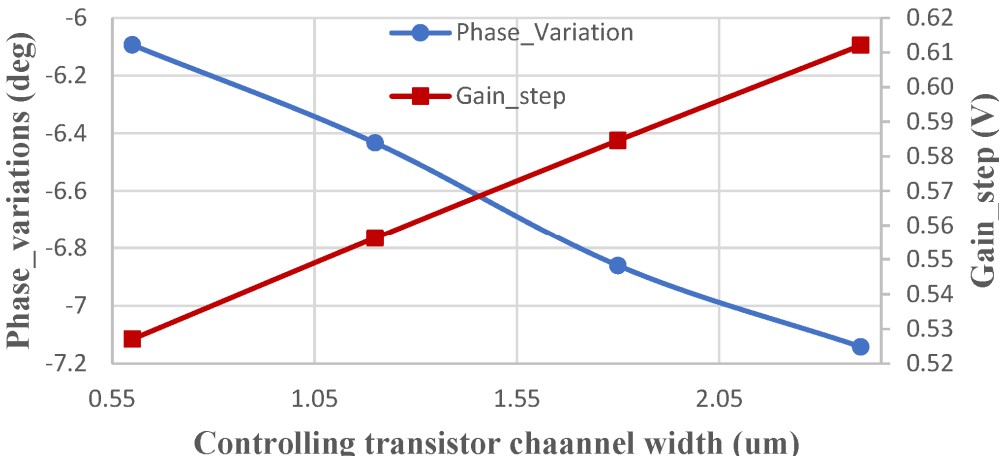

**Figure 10.** Phase variations and gain step versus the controlling transistor channel width.

**Table 1.** Specifications and the corresponding parameters in the control.

| Controlling Parameters | Specifications | | | |
|---|---|---|---|---|
| | **Gain Step** | **Phase Variation** | **Stability** | **Linearity** |
| Controlling transistors sizes ($W_{controlling}$) | ✓ | | | |
| Output matching network ($C_o$, $L_{ct}$, $L_{o1}$, and $L_{op}$) | ✓ | ✓ | | ✓ |
| Interstage inductor ($L_{int}$) | | ✓ | | |
| Transistor sizing ($W_m$ and $W_c$) | | | ✓ | ✓ |
| RC—feedback network ($C_F$ and $R_F$) | | | ✓ | |
| Source degenerated inductor ($L_s$) | ✓ | | | ✓ |
| DC source ($I_{dc}$) | ✓ | | | ✓ |

### 3.2.2. Knowledge-Aware Simulation-Based Optimization Flow

The proposed optimization algorithm depends on the optimizer core algorithm (BFGS) built within the virtuoso environment, and the proposed knowledge-aware and the weighting factor methodologies.

The knowledge-aware methodology divides the complex optimization problem into several smaller optimization steps. Those steps depend on the knowledge and experience of the designer to identify the sensitivity of the performance specifications to the design parameters. The studied sensitivity test is done through deriving analytical expressions and is verified by simulations. Table 2 illustrates the different optimization steps to speed up the optimization process. It includes the optimization steps, the corresponding performance specification, and the design parameters to optimize.

Weighting factors are used for the different goals to help the optimizer to converge and reach an optimum solution that satisfies the different specified constraints. These weighting factors are entered by the user into the input interface of the proposed tool according to his priority list of specifications.

**Table 2.** The optimizer different stages with the corresponding parameters to be optimized for the required specification.

| Steps | Targeted Specification | Parameters to Be Optimized |
|---|---|---|
| 1 | S11 | I/P matching network elements ($L_{mi}$, $C_{in}$, $C_{ms}$, and $C_{ml}$) |
| 2 | S21 | I/P matching network elements ($L_{mi}$, $C_{in}$, $C_{ms}$, and $C_{ml}$) and O/P matching network elements ($L_o$, $L_{o1}$, $L_{op}$, $C_o$, and $C_{op}$) |
| 3 | Gain Step and RMS gain error | Sizing of the controlling transistors ($W_{con}$) |
| 4 | RMS Phase error | Interstage inductor ($L_{int}$) and O/P matching network elements ($L_o$, $L_{o1}$, $L_{op}$, $C_o$, and $C_{op}$) |
| 5 | Linearity (IP1dBc and IIP3) | Source degenerated inductor ($L_s$), Transistor sizing ($W_m$), $I_{dc}$, and O/P matching network elements ($L_o$, $L_{o1}$, $L_{op}$, $C_o$, and $C_{op}$) |
| 6 | Stability and Bandwidth realization | F.B. RC network elements (Rcas and Ccas) and the inductor of the output matching network ($L_{o1}$) |

The RF simulator generates the inclusive netlist and then invokes the optimizer using the initial design parameters saved within the tool. The optimizer starts its optimization algorithm. After each optimization stage, according to Table 2, the inclusive netlist is updated by the results of each optimization stage. Below is a summary of the design flow:

1. The user enters the required specifications with their weighting factors through the web GUI, while initial design parameters are available within the tool.
2. The spice simulator generates the schematic inclusive netlist from virtuoso and the parasitic device model generator.
3. The optimizer starts its optimization algorithm (BFGS) guided by the weighting factors entered by the user to know which design specification to meet first.
4. Firstly, it checks on the transistors' regions making sure that transistors are in their correct region. If this check fails, the optimizer will sweep on the transistor's sizes and the dc current source till it passes. Every optimization loop will be on a more confined range for the concerned design parameters.
5. The optimizer then simulates all the specifications giving a message indicating which specification passes and which fails.
6. Depending on the weighting factor optimization methodology, the optimizer chooses which specification to meet first. Following table II, shown below, the optimizer will enter the right step stage and thereby sweeps on the parameters in favor of the specified specification.
7. Repeat steps 4 to 6 till all the targeted specifications are achieved.
8. Finally, the tool displays the optimized parameters, the achieved specifications, the circuit schematic, and the template of the layout with the GDSII file for further modification.

Figure 11 shows a comprehensive flow chart describing the optimization algorithm. Additionally, Table 2 shows the different step stages the optimizer can enter with the corresponding parameters to optimize on for the specification needed to be met.

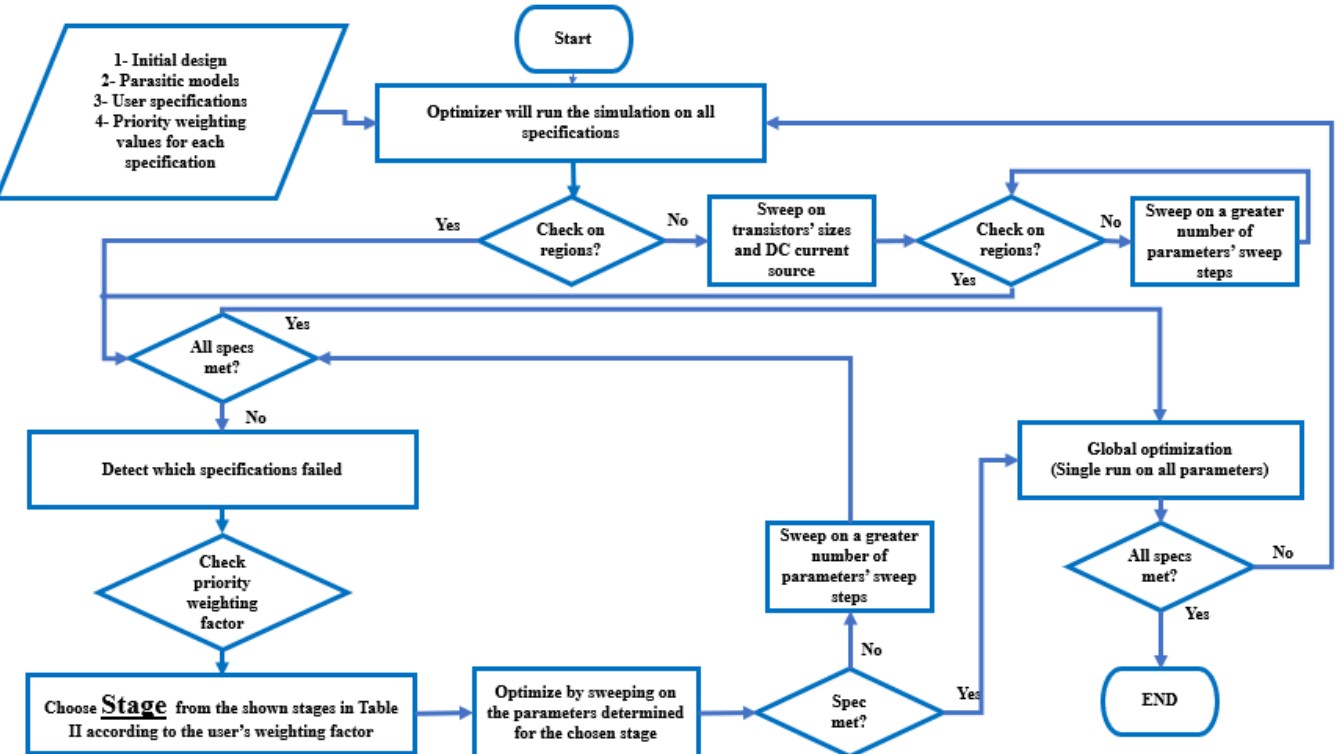

**Figure 11.** Comprehensive optimization flow chart for the presented optimizer.

## 4. Verification and Simulation Results

*Simulation Results*

The proposed tool is used for the automatic sizing of the current steering VGAs. It shows optimal design performance on a range of frequencies up to 15 GHz. The design of the VGA, using the proposed tool, is done at 7 GHz, 10 GHz, and 13 GHz using a 65nm-technology node. Table 3 shows the initial design parameters for any design specifications. In addition, a template for the layout is generated to help the user to finalize the layout. For the three test frequencies, Figure 12 shows the simulated phase and gain versus the number of states, respectively. Figures 13 and 14 show the simulated $S_{21}$ and $S_{11}$ versus frequency, respectively. As depicted in the graphs and Table 4, the requested specifications for the three test frequencies are achieved.

**Table 3.** Initial design parameter values for the optimizer.

| Input Matching Network | | Output Matching Network | | Intermediate Inductance | |
|---|---|---|---|---|---|
| $C_{in}$ (fF) | 850 | $L_{ct}$ (pH) | 600 | $L_{int}$ (pH) | 700 |
| $C_{ml}$ (fF) | 700 | $L_{o1}$ (pH) | 600 | **Cascode RC network** | |
| $C_{ms}$ (fF) | 700 | $L_{op}$ (pH) | 600 | Rcas (KΩ) | 10 |
| $L_{mi}$ (pH) | 400 | $C_{mo}$ (fF) | 400 | Ccas (fF) | 80 |
| **Channel width for RF CMOS transistors** | | | | **Feedback network** | |
| $W_m = W_c = W_p$ (um) | 2 | $N_{con3} = N_{p3}$ | 8 | RF (KΩ) | 1 |
| $W_{con}$ (nm) | 600 | $N_{con4} = N_{p4}$ | 4 | CF (fF) | 80 |
| $N_m = N_c$ | 32 | $M_m$ | 2 | **Source degenerated inductance** | |
| $N_{con1} = N_{p1}$ | 32 | $M_c$ | 1 | Ls (pH) | 300 |
| $N_{con2} = N_{p2}$ | 16 | $M_{con}$ | 1 | | |

where m: main, C: cascode, con: controlling, p: parallel, W: width per finger, N: number of fingers, and M: number of multipliers.

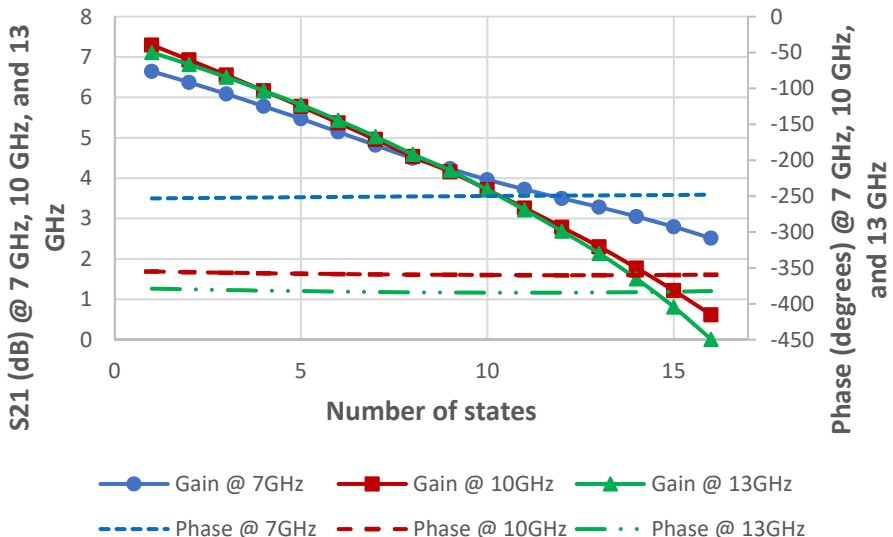

**Figure 12.** S21 and Phase versus number of states at the three test frequencies (7, 10, and 13 GHz).

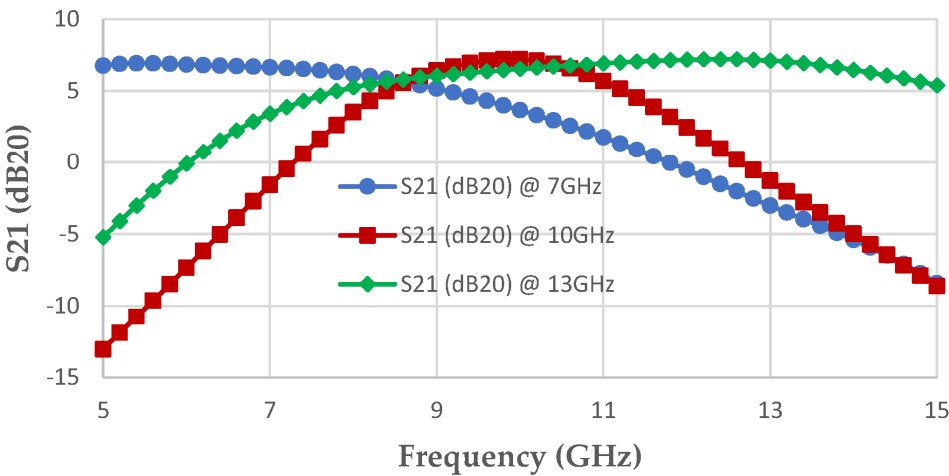

**Figure 13.** Simulated S21 versus frequency at 7 GHz, 10 GHz, and 13 GHz.

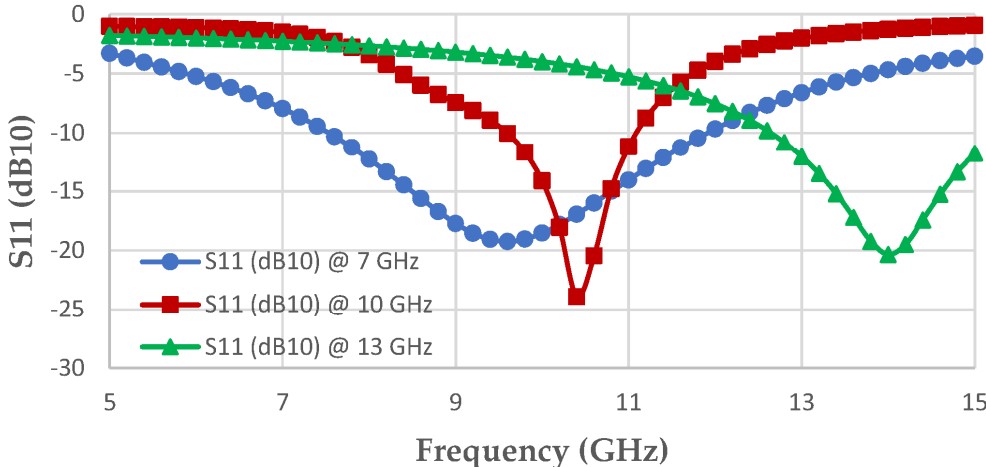

**Figure 14.** Simulated S11 versus frequency at 7 GHz, 10 GHz, and 13 GHz.

**Table 4.** Comparison between the achieved specifications from the proposed tool at the three test frequencies and the required ones.

| Specifications | Required Specifications at 7 GHz | Achieved Specifications at 7 GHz | Required Specifications at 10 GHz | Achieved Specifications at 10 GHz | Required Specifications at 13 GHz | Achieved Specifications at 13 GHz |
|---|---|---|---|---|---|---|
| $S11_{Frf}$ | $<-7$ dB | $-8$ dB | $<-7$ dB | $-14.11$ dB | $<-7$ dB | $-12.05$ dB |
| $S21_{max}$ | $>5$ dB | $6.64$ dB | $>5$ dB | $7.2$ dB | $>5$ dB | $7.11$ dB |
| $IP1dB_{min}$ | $>-3$ dBm | $5.2$ dBm | $>-3$ dBm | $0.83$ dBm | $>-3$ dBm | $3.7$ dBm |
| Bandwidth-min | 3 GHz | 3 GHz | 1.5 GHz | 1.7 GHz | 3 GHz | 4 GHz |
| Phase Variation | $(-5°)$–$5°$ | $-5°$ | $(-5°)$–$5°$ | $4.49°$ | $(-5°)$–$5°$ | $3.295°$ |
| Gain Step | 0.3 dB | 0.2581 dB | 0.5 dB | 0.417 dB | 0.4 dB | 0.3367 dB |
| IIP3 | $>3$ dB | 12.67 dBm | $>3$ dB | 8.6 dBm | $>3$ dB | 11.95 dBm |
| RMS Gain Error | 0.1–0.2 | 0.05 | 0.2–0.3 | 0.085 | 0.1–0.3 | 0.035 |
| RMS Phase Error | $0.2°$–$0.3°$ | $0.34°$ | $0.2°$–$0.3°$ | $0.26°$ | $0.4°$–$0.6°$ | $0.6°$ |

The high current of RF circuits puts high constraint on the devices' layout. Electromigration is considered to avoid device breakdown. The circuit's layout was made for the 10 GHz design as a template layout for other designs. The layout took an active area 960 μm × 1090 μm, as shown in Figure 15.

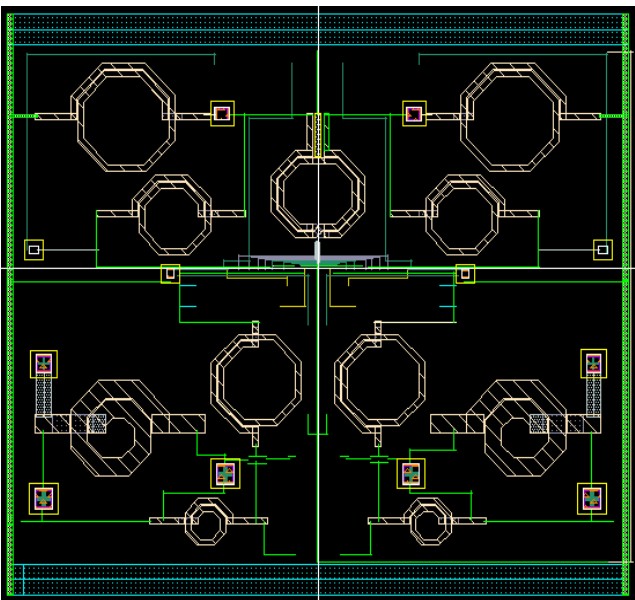

**Figure 15.** Full layout for the presented VGA at 10 GHz.

## 5. Conclusions

Web-based parasitic-aware automation and optimization RF design tool for mm-wave digitally controlled current steering VGAs has been proposed in this paper. The tool uses an optimizer embedded in a virtuoso environment, which uses the BFGS algorithm to reach optimal design parameters for the users' targeted specifications. The theory and the optimization algorithm were demonstrated in this paper. The tool considers before design procedures the parasitics of both the active and the passive devices. MOSFETs' parasitics and its interconnections are modeled as physical ideal capacitances, and analytical equations for the estimated parasitic capacitances are obtained through a linear regression method which relates the parasitic capacitance to the transistor sizes. Inductors and capacitors are used from the technology design kit, 65nm technology node, for parasitic inclusion. The obtained results for the three test frequencies assure the effectiveness of the parasitic aware design technique proposed in the presented tool.

**Author Contributions:** Conceptualization, N.M. and M.E.; formal analysis, N.M.; investigation, N.M.; methodology, N.M.; software, N.M.; supervision, H.R.; validation, N.M.; visualization, N.M.; writing—original draft, N.M.; writing—review and editing, M.E. All authors have read and agreed to the published version of the manuscript.

**Funding:** This research received no external funding.

**Data Availability Statement:** Some of the data presented in this study are available on request from the corresponding author.

**Acknowledgments:** The authors would like to thank engineer Ahmed Samir, for his genuine help and support in the coding part of the web interface.

**Conflicts of Interest:** The authors declare no conflict of interest.

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
