# Peer review of "Parasitic-Aware Simulation-Based Optimization Design Tool for Current Steering VGAs"

_electronics, doi:10.3390/electronics12010132_

Round 1

Reviewer 1 Report

1.     The authors present web-based parasitic-aware automation and optimization design tool for digitally con-11 trolled current steering VGAs. The optimization is based on a simulator based optimizer embedded within the virtuoso environment.

2.     In the table II, the optimizer different stages with the corresponding parameters to be optimized for the required specification should be demonstrated in detail.

3.     The manuscript has 20 figures; the number of the figures should be decreased.

4.     Revise the English thoroughly before submission.

Reviewer 2 Report

The authors have used PAS-based optimization design tool for current steering VGAs. This paper is well-written but need some minor improvements. Following are my comments.

1. Abstract is short. Add more quantitative analysis and potential applications.

2. The novelty of this work should be more stressed in last paragraph of the introduction section.

3. Quality of Fig. 1 should be improved. 

4. In Fig. 1, RC FB Network is not properly explained. Explain this in details.
